# Mean-field model of interacting quasilocalized excitations in glasses

Corrado Rainone[1], Pierfrancesco Urbani[2], Francesco Zamponi[3],
Edan Lerner[1⋆] and Eran Bouchbinder[4†]

1 Institute for Theoretical Physics, University of Amsterdam,
Science Park 904, Amsterdam, Netherlands
2 Université Paris-Saclay, CNRS, CEA, Institut de physique théorique,
91191, Gif-sur-Yvette, France
3 Laboratoire de Physique de l'Ecole Normale Supérieure, ENS, Université PSL, CNRS,
Sorbonne Université, Université de Paris, F-75005 Paris, France
4 Chemical and Biological Physics Department,
Weizmann Institute of Science, Rehovot 7610001, Israel

⋆ e.lerner@uva.nl,   † eran.bouchbinder@weizmann.ac.il

## Abstract

Structural glasses feature quasilocalized excitations whose frequencies $\omega$ follow a universal density of states $\mathcal{D}(\omega) \sim \omega^4$. Yet, the underlying physics behind this universality is not fully understood. Here we study a mean-field model of quasilocalized excitations in glasses, viewed as groups of particles embedded inside an elastic medium and described collectively as anharmonic oscillators. The oscillators, whose harmonic stiffness is taken from a rather featureless probability distribution (of upper cutoff $\kappa_0$) in the absence of interactions, interact among themselves through random couplings (characterized by a strength $J$) and with the surrounding elastic medium (an interaction characterized by a constant force $h$). We first show that the model gives rise to a gapless density of states $\mathcal{D}(\omega) = A_g \omega^4$ for a broad range of model parameters, expressed in terms of the strength of the oscillators' stabilizing anharmonicity, which plays a decisive role in the model. Then — using scaling theory and numerical simulations — we provide a complete understanding of the non-universal prefactor $A_g(h, J, \kappa_0)$, of the oscillators' interaction-induced mean square displacement and of an emerging characteristic frequency, all in terms of properly identified dimensionless quantities. In particular, we show that $A_g(h, J, \kappa_0)$ is a non-monotonic function of $J$ for a fixed $h$, varying predominantly exponentially with $-(\kappa_0 h^{2/3}/J^2)$ in the weak interactions (small $J$) regime — reminiscent of recent observations in computer glasses — and predominantly decays as a power-law for larger $J$, in a regime where $h$ plays no role. We discuss the physical interpretation of the model and its possible relations to available observations in structural glasses, along with delineating some future research directions.

## 1 Introduction

Many key mechanical, dynamic and thermodynamic phenomena in structural glasses — ranging from wave attenuation and heat transport to elasto-plastic deformation and yielding — are controlled by the abundance and micromechanical properties of low-frequency (soft) quasilocalized vibrational modes (QLMs) [1–7]. These nonphononic excitations (see example in Fig. 1a) emerge from self-organized glassy frustration [8], which is generic to structural glasses quenched from a melt [9]. Their associated frequencies $\omega$ have been shown [10–12] to follow a universal nonphononic (non-Debye) density of states $\mathcal{D}(\omega) \sim \omega^4$ as $\omega \to 0$, independently of microscopic details [13, 14], spatial dimension [15, 16] and formation history [17, 18]. Some examples for $\mathcal{D}(\omega)$, obtained in computer glasses, are shown in Fig. 1b. Due to the prime importance of soft QLMs for many aspects of glass physics, developing theoretical understanding of their emergent statistical-mechanical properties is a timely challenge.

Nearly two decades ago, Gurevich, Parshin and Schober (GPS) put forward a three-dimensional ($\eth = 3$, where $\eth$ is the spatial dimension) lattice model [19], aimed at resolving the vibrational density of states of QLMs. The model assumes QLMs to exist inside an embedding elastic medium and to be described as anharmonic oscillators — meant to represent small, spatially-localized sets of particles — that are characterized by a stiffness probability distribution $p(\kappa)$ in the absence of interactions. The oscillators interact with each other via random couplings, which are characterized by a strength $J$ that follows the $\sim r^{-\eth}$ spatial decay of linear-elastic dipole-dipole interactions, where $r$ is the distance between the oscillators. GPS showed numerically that the model's vibrational spectrum indeed grows from zero frequency as $\mathcal{D}(\omega) \sim \omega^4$ [19] for various choices of $p(\kappa)$, and these numerical results have been rationalized using a phenomenological theory [19–21].

Yet, despite previous efforts [1, 22–26], we currently lack insight into the origin of QLMs' statistical-mechanical properties. Moreover, recent progress in studying computer glass-formers revealed intriguing properties of QLMs [17, 18, 27], e.g. the dependence of the amplitude $A_{\mathrm{g}}$ of the $\omega^4$ universal law on the state of glassy disorder (cf. inset of Fig. 1b), which are not yet fully understood. In this work we study — using scaling theory and numerical simulations — the spectral properties of the mean-field variant of GPS's lattice model, obtained by taking the limit of infinite spatial dimension ($\eth \to \infty$), and by allowing the oscillators to also interact with their surrounding elastic medium through a constant force $h$. A similar mean-field model, albeit without a force term, was studied by Kühn and Horstmann [28] in the context of low-temperature glassy anomalies [29–31]. We therefore refer to our model hereafter as

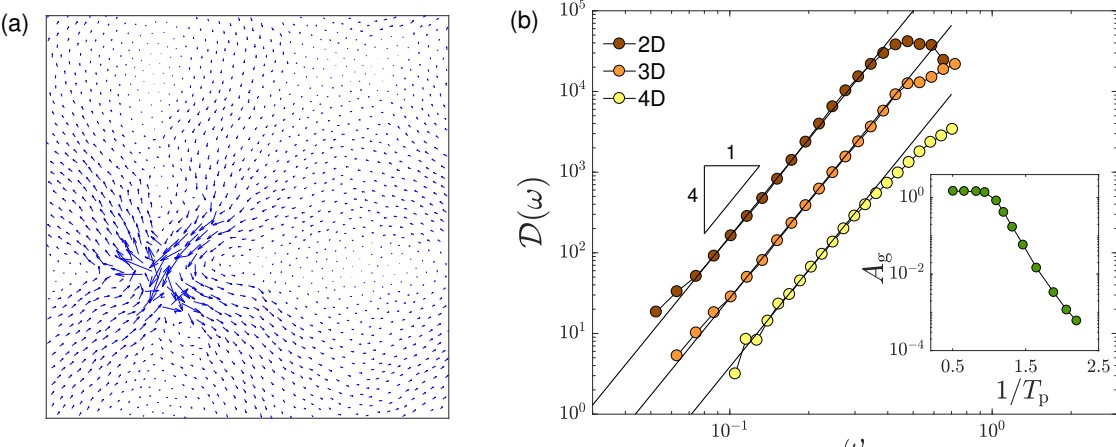

Figure 1: (a) An example of a QLM observed in a two-dimensional (2D) computer glass [15]. (b) Nonphononic spectra $\mathcal{D}(\omega)$ vs. frequency $\omega$, reported for 2D, 3D and 4D soft-sphere computer glasses [15]. Data are shifted vertically for visual clarity. (inset) The prefactor $A_{\mathrm{g}}$ of the nonphononic $\omega^4$ spectrum vs. the inverse of the parent equilibrium temperature $T_{\mathrm{p}}$ from which glasses were instantaneously quenched, calculated for the soft-sphere computer glass model investigated in [17].

the KHGPS model.

In this work, we show that the low-frequency spectrum of the KHGPS model[1] — to be explicitly formulated below — rather generically follows $\mathcal{D}(\omega) = A_{\mathrm{g}} \, \omega^4$, as is widely observed in particle-based computer glass-formers, cf. Fig. 1b. Furthermore, we develop a complete understanding of the non-universal prefactor $A_{\mathrm{g}}(h, J, \kappa_0)$ (where $\kappa_0$ characterizes the initial stiffness probability distribution $p(\kappa)$, see details below), of the oscillators' interaction-induced mean square displacement, and of an emerging characteristic frequency, all in terms of properly identified dimensionless quantities. In particular, we show that $A_{\mathrm{g}}(h, J, \kappa_0)$ is a non-monotonic function of $J$ for a fixed $h$ and strength of anharmonicity, varying predominantly according to $\log[A_{\mathrm{g}}(h, J, \kappa_0)] \sim -(\kappa_0 h^{2/3}/J^2)$ in the weak interactions (small $J$) regime — reminiscent of recent observations in computer glasses shown in the inset of Fig. 1b — and predominantly decays as a power-law for larger $J$, in a regime where $h$ plays no role. We discuss the physical interpretation of the model and its possible relations to available observations in structural glasses, along with delineating some future research directions.

## 2 The model

QLMs in glasses have been shown to feature large displacements inside a localized core of a few atomic distances in linear size, accompanied by power-law decaying dipolar displacements away from the core (cf. Fig. 1a). QLMs also feature low vibrational frequencies, i.e. they represent particularly soft regions inside a glass, and are randomly distributed in space. The main question we aim at addressing in this work is whether one can develop a relatively simple mean-field model, using this physical picture of QLMs as an input, to obtain their universal density of states $\mathcal{D}(\omega) = A_{\mathrm{g}} \, \omega^4$ and to gain insight into the properties of the non-universal prefactor $A_{\mathrm{g}}$.

---

[1]We study zero temperature states (local minima) reached by a minimization of the Hamiltonian, starting from random states. The spectrum of true ground states of the model might be different.

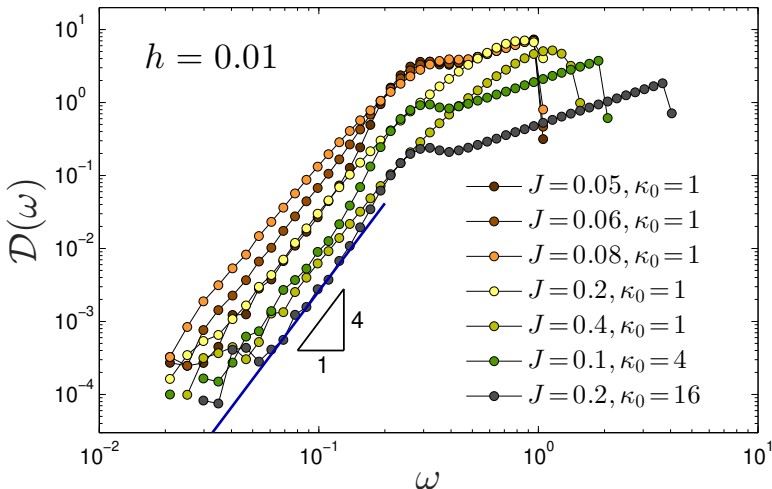

Figure 2: The vibrational density of states of the KHGPS model, calculated numerically in systems of $N = 16000$ oscillators, for $h = 0.01$ and various values of the parameters $J$ and $\kappa_0$ as indicated by the legend, see text for further discussion. The linear scaling at high frequencies seen in some regimes of parameter-space is a remnant of the initial flat distribution of oscillators' harmonic stiffnesses $p(\kappa) = \kappa_0^{-1}$.

To this aim, we closely follow GPS and adopt a coarse-grained picture in which QLMs are anharmonic oscillators embedded inside an elastic medium. The elastic medium mediates interactions between the QLMs and can also affect them directly. We consider a collection of $N$ anharmonic oscillators, each described by a generalized coordinate $x_i$, whose Hamiltonian takes the form

$$H \equiv \frac{1}{2}\sum_i \kappa_i x_i^2 + \frac{A}{4!}\sum_i x_i^4 + \sum_{i<j} J_{ij} x_i x_j - h\sum_i x_i \,. \tag{1}$$

The oscillators in Eq. (1) are characterized by random harmonic stiffnesses $\kappa_i$, extracted from a probability distribution $p(\kappa)$ (see details and discussion below). They also feature a fourth order stabilizing anharmonicity of strength $A$[2]. Each anharmonic oscillator is coupled to all other oscillators by interaction coefficients $J_{ij}$, assumed to be Gaussian, i.i.d. random variables of variance $J^2/N \ \forall \ i \neq j$. As the anharmonic oscillators are thought to be embedded inside an elastic medium, $J$ represents the strength of disorder in the emerging elastic interactions, taken to be space-independent in the mean-field framework. Finally, the elastic medium generically features internal forces that act on the oscillators, mimicked by a constant field $h$ that is linearly coupled to the generalized coordinates $x_i$. Note that $h$ breaks the $x_i \to -x_i$ symmetry of the Hamiltonian.

## The low-frequency density of states

Our first goal is to understand whether, and if so under what conditions, the KHGPS Hamiltonian in Eq. (1) with rather featureless initial distributions $p(\kappa)$ leads to $\mathcal{D}(\omega) \sim \omega^4$, where $\omega^2$ is the stiffness characterizing the minima of $H$. More formally, we are interested in the spectrum of the Hessian, $\mathcal{M}_{ij} \equiv \partial^2 H/\partial x_i \partial x_j = J_{ij} + \delta_{ij}(\kappa_i + \frac{1}{2}A x_i^2)$, evaluated at positions $(x_*)_i$ for which $H$ attains a minimum. The off-diagonal contribution, $J_{ij}$, represents a Gaus-

---

[2]We set $A$ to be (the same) constant for all oscillators, consistent with direct calculations for soft, localized modes in computer glasses [10, 32]

sian random matrix that does not give rise to an $\omega^4$ spectrum [33]. The diagonal contribution, which potentially gives rise to an $\omega^4$ spectrum, is a sum of $\kappa_i$ that follows an input distribution $p(\kappa)$ and of $\frac{1}{2}Ax_i^2$. The statistics of the latter, corresponding to the stabilizing anharmonicity, is therefore the most important part. Note that while we focus on studying the zero temperature $(T \to 0)$ properties of the model, i.e. the statistical properties of the Hessian matrix $\mathcal{M}_{ij}$, we envision that $p(\kappa)$ encodes information about a glass-forming liquid above its glass transition temperature, and that the minimization of $H$ mimics the self-organization processes the liquid undergoes while quenched to a low $T$ during glass formation.

Since instantaneous liquid states typically feature also negative stiffnesses [34–36], we expect $p(\kappa = 0) \geq 0$. For simplicity, we take this expectation into account by considering $p(\kappa) = \kappa_0^{-1}$. That is, we hereafter take $p(\kappa)$ to correspond to a uniform probability distribution over the interval $[0, \kappa_0]$, where $\kappa_0$ is a stiffness scale characterizing the liquid state in which the oscillators are taken to be non-interacting. As the temperature is reduced during a quench, elasticity builds up and finite interactions emerge (i.e. finite $J_{ij}$ and $h$). The latter restructure the initial distribution $p(\kappa)$ into $\mathcal{D}(\omega)$, characterizing the ensemble of minima of $H$. In the language of GPS [19, 20], $p(\kappa)$ undergoes complete reconstruction well below a frequency scale $\omega_\times$ (to discussed below) upon minimizing $H$.

We measure $x_i$ relative to some reference position $x_0$, which is also taken to set the unit length in the model. Energy is measured in units of $Ax_0^4$. Consequently, $\kappa_i$, $\kappa_0$ and $J$ are measured in units of $Ax_0^2$, and the force $h$ in units of $Ax_0^3$. We first study the KHGPS model numerically by initializing $N = 16000$ oscillators placed at $x_i = 0$, and assigning values for the parameters $(h, J, \kappa_0)$. As explained above, we draw $\kappa_i$ from a uniform distribution over the interval $[0, \kappa_0]$ and the couplings $J_{ij}$ from a Gaussian distribution of width $J/\sqrt{N}$. We then minimize the Hamiltonian given in Eq. (1) with respect to the coordinates $x_i$ by a standard nonlinear conjugate gradient minimization. The Hessian $\mathcal{M}_{ij}$ is evaluated and diagonalized upon reaching a minimum, where the oscillators attain new displacements $(x_*)_i$. This procedure is repeated at least 1150 times for each $(h, J, \kappa_0)$, and the statistics of the respective spectra are analyzed.

The resulting density of states $\mathcal{D}(\omega)$, for a rather broad range of parameter sets $(h, J, \kappa_0)$, are shown in Fig. 2. It is observed that in all cases there exists a low-frequency regime in which $\mathcal{D}(\omega) \sim \omega^4$. We take this numerical evidence to indicate that the KHGPS model features a gapless density of states $\mathcal{D}(\omega) = A_g \omega^4$ for a broad range of model parameters. The theoretical status of this statement is further discussed below. Next, taking $\mathcal{D}(\omega) \sim \omega^4$ to be generically valid, we shift our focus to the dependence of the main emergent quantities in the model on the parameters $h$, $J$ and $\kappa_0$. In particular, we aim at obtaining a theoretical understanding of the characteristic frequency scale $\omega_\times$, of the mean square displacement $\langle x_*^2 \rangle$ of the oscillators at minima of $H$ and of the prefactor $A_g$.

The frequency scale $\omega_\times$ divides the initial frequency domain $0 \leq \omega \leq \omega_0$ (with $\omega_0^2 \equiv \kappa_0$) into a low frequency regime that undergoes reconstruction and a high frequency regime that does not. Indeed, $\mathcal{D}(\omega)$ is observed to vary linearly with $\omega$ in the high frequency regime of Fig. 2 — corresponding to the initial uniform $p(\kappa)$ —, where $\mathcal{D}(\omega) \sim \omega^4$ emerges at significantly smaller frequencies. $\langle x_*^2 \rangle$ is the average of the $x_i$ dependent part of the Hessian $\mathcal{M}_{ij} = J_{ij} + \delta_{ij}(\kappa_i + \frac{1}{2}Ax_i^2)$ at minima of $H$, and quantifies the average interaction-induced force that the oscillators generate, as will be further discussed below. Finally, $A_g(h, J, \kappa_0)$ is a non-universal quantity that — in structural glasses — encodes information about the non-equilibrium history of the material, having fundamental implications for its physical properties [17, 37, 38].

The approach we take aims at developing a comprehensive scaling theory of the KHGPS model, identifying the main quantities that control its behavior, the relevant groups of dimensionless parameters and the different regimes it exhibits. The scaling predictions are then

being quantitatively tested against extensive numerical simulations of the model. Such an approach provides valuable insight into the possible relations between the model and realistic glasses, most notably the model's potential implications for our understanding of quasilocalized excitations in glasses, including their universal and history-dependent properties.

# 3   The weak interactions regime

We first consider the weak inter-oscillator interactions regime, i.e. situations in which the force $h$ is finite and $J$ is small (note that $h$ and $J$ have different physical units, so this statement should be properly recast in dimensionless form, as will be done below). Our strategy is to first understand the properties of the oscillators in the non-interacting case, $J = 0$, and then to treat the effect of small $J > 0$ perturbatively. In the non-interacting case, $J = 0$, the single oscillator Hamiltonian $H_s$ takes the form $H_s = \frac{1}{2}\kappa x^2 + \frac{1}{4!}x^4 - h x$ (note that here $A$ and $x_0$ are already used to set the units of all of the other quantities). We expect $\mathcal{D}(\omega) \sim \omega^4$ to emerge for $\omega \ll \omega_\times$ upon the introduction of interactions, $J > 0$, but also expect $\omega_\times$ itself not to be affected by $J$ in the weak interactions regime. Consequently, $\omega_\times$ is determined by the oscillations frequency at the minimum of $H_s$ for small $\kappa$, i.e. $\omega_\times(h) \sim h^{1/3}$.

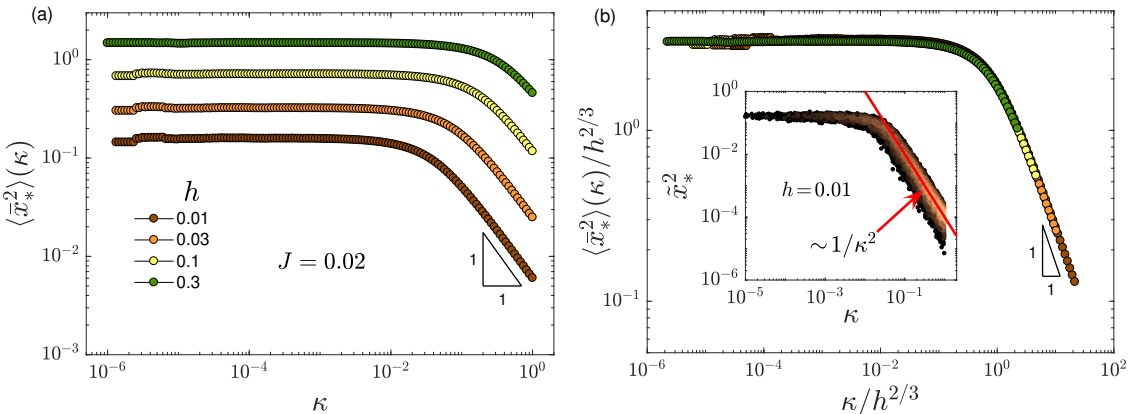

Figure 3: (a) The partial averages $\langle \bar{x}_*^2 \rangle(\kappa)$ of oscillators' squared displacements (see text for precise definitions) in the weak interactions regime, summed over and plotted against $\kappa$, for various strengths of the field $h$ as specified in the legend, and $\kappa_0 = 1$. (b) Rescaling $\langle \bar{x}_*^2 \rangle(\kappa)$ and $\kappa$ by $\omega_\times^2 \sim h^{2/3}$ leads to a perfect collapse of the partial averages, validating our prediction $\omega_\times \sim h^{1/3}$ in the weak interaction regime. Inset: scatter-plotting $\tilde{x}_*^2$ vs. $\kappa$ validates the predicted $\sim \kappa^{-2}$ scaling for $\kappa \gtrsim \omega_\times^2$, see text for discussion.

The basic roles played by the frequency scale $\omega_\times$ in the model can be further demonstrated by considering the oscillators' mean square displacement $\langle x_*^2 \rangle$ ($\langle \bullet \rangle$ stands for averaging over the statistics of both $\kappa$ and $J_{ij}$). Let us consider the displacement $\tilde{x}_*$ of oscillators featuring $\sqrt{\kappa} \ll \omega_\times(h)$. That is, $\tilde{x}_*$ is the displacement of individual oscillators as a function of their initial stiffness $\kappa$. Analyzing $H_s$ in this limit, assuming that $J$ makes a negligible contribution to $\tilde{x}_*$, leads to $\langle \tilde{x}_*^2 \rangle(\kappa) \sim \omega_\times^2$, i.e. $\langle \tilde{x}_*^2 \rangle(\kappa)$ is predicted to be independent of $\kappa$ in this limit. Considering the opposite limit, $\sqrt{\kappa} \gg \omega_\times(h)$, we obtain $\langle \tilde{x}_*^2 \rangle(\kappa) \sim \kappa^{-2}$. Note that the existence of a frequency domain $\sqrt{\kappa} \gg \omega_\times(h)$ implies that $\kappa_0$ — the upper cutoff of $p(\kappa)$ — is in fact the largest stiffness scale in the problem (compared to $J$ and $h^{2/3}$, which are also of stiffness dimensions). To see how $\langle x_*^2 \rangle$ emerges from the $\kappa$-dependent $\langle \tilde{x}_*^2 \rangle(\kappa)$, we define the partial average $\langle \bar{x}_*^2 \rangle(\kappa) = \kappa^{-1} \int_0^\kappa \langle \tilde{x}_*^2 \rangle(\kappa') d\kappa'$. The motivation for defining the partial average is that

$\langle x_*^2 \rangle = \langle \bar{x}_*^2 \rangle (\kappa_0)$ (recall that $p(\kappa) = \kappa_0^{-1}$), i.e. it provides insight into the statistical weight of the different $\kappa$ regimes in the emerging $\langle x_*^2 \rangle$. Evaluating the partial average for $\sqrt{\kappa} \ll \omega_\times(h)$, we obtain $\langle \bar{x}_*^2 \rangle (\kappa) \sim \omega_\times^2 \sim h^{2/3}$, while for $\sqrt{\kappa} \gg \omega_\times(h)$ we have $\langle \bar{x}_*^2 \rangle (\kappa) \sim \omega_\times^4 / \kappa \sim h^{4/3} / \kappa$, where the amplitude of the latter has been set such that the two scaling laws smoothly connect at $\sqrt{\kappa} \approx \omega_\times(h)$. The scaling predictions for $\langle \bar{x}_*^2 \rangle (\kappa)$ are fully supported by numerical simulations, as shown in Fig. 3. Consequently, the mean square displacement in the weak interactions regime is predicted to follow $\langle x_*^2 \rangle \sim h^{4/3} / \kappa_0$, to be verified later.

Up until now, the interaction strength $J$ has not appeared explicitly in the quantities discussed, though $J > 0$ is essential for having $\mathcal{D}(\omega) \sim \omega^4$. How does $J$ enter the problem in the weak interactions regime? To start addressing this question, we first ask what is the dimensionless combination of parameters in which a small $J$ can appear. Since $J$ has the dimension of stiffness (i.e. frequency squared) and since $\omega_\times(h)$ is a relevant $J$-independent frequency scale in the problem, we expect the smallness of $J$ to be manifested through the ratio $J / \omega_\times(h)$. The latter, which is of frequency dimension, can be made dimensionless using the other large frequency scale in the problem, i.e. $\sqrt{\kappa_0}$. Consequently, a scaling consideration predicts that $J$ enters the problem in the weak interactions regime through the dimensionless combination $y \equiv J / (h^{1/3} \kappa_0^{1/2})$.

To understand the appearance of $J$ in the weak interactions regime, we consider next $A_g(h, J, \kappa_0)$, the non-universal prefactor of the universal $\sim \omega^4$ density of states. To obtain a scaling estimate of $A_g(h, J, \kappa_0)$, we note that the number of oscillators that undergo interaction-induced reconstruction is $\sim (\omega_\times^2 / \kappa_0) N$, just by the definition of $\omega_\times$. If the upper frequency cutoff of the density of states $\mathcal{D}(\omega) = A_g(h, J, \kappa_0) \omega^4$ is proportional to $\omega_\times$, we then obtain $\int_0^{\omega_\times} \mathcal{D}(\omega) d\omega \sim (\omega_\times^2 / \kappa_0) N$. The latter implies $A_g(h, J, \kappa_0) = g(y) / (\kappa_0 [\omega_\times(h)]^3) = g(y) / (\kappa_0 h)$, where $g(y)$ is a dimensionless function of the small dimensionless quantity $y = J / (h^{1/3} \kappa_0^{1/2}) \ll 1$, which cannot be obtained by pure scaling considerations.

In order to go beyond pure scaling theory, one needs to invoke an effective description of the full Hamiltonian of Eq. (1). That is, one may ask how the interactions with all the other oscillators — characterized by the couplings $J_{ij}$ — affect an effective oscillator of stiffness $\kappa$. In the most general case, interactions shift $\kappa$ by an amount denoted by $\kappa_{\text{shift}}(h, J, \kappa_0)$, and generate an effective force $f(h, J, \kappa_0)$ in addition to $h$ [19, 20]. Consequently, a representative oscillator of stiffness $\kappa$ and position $x$ is described by an effective potential of the form

$$v_{\text{eff}}(x) = [\kappa - \kappa_{\text{shift}}(h, J, \kappa_0)] \frac{x^2}{2} + \frac{x^4}{4!} - [h + f(h, J, \kappa_0)] x . \tag{2}$$

Comparing Eq. (1) to Eq. (2), we immediately conclude that the $\sum_{i<j} J_{ij} x_i x_j$ term in the former corresponds to the $f(h, J, \kappa_0) x$ term in the latter, where $f(h, J, \kappa_0)$ is normally distributed, with a zero mean and a standard deviation of $J \sqrt{\langle x_*^2 \rangle}$, for sufficiently large $N$ (according to the central limit theorem). This result yet again demonstrates the importance of the mean square displacement $\langle x_*^2 \rangle$. Obtaining the effective shift $\kappa_{\text{shift}}(h, J, \kappa_0)$ is more involved; at this point, we assume it is negligible in the weak interaction limit, an assumption that will be validated a posteriori below.

To obtain the dimensionless function $g(y)$ in $A_g(h, J, \kappa_0) = g(y) / (\kappa_0 h)$, we consider Eq. (2) with $\kappa_{\text{shift}}(h, J, \kappa_0) = 0$. When $f(h, J, \kappa_0) = 0$, i.e. in the non-interacting case discussed above, oscillators in the initial frequency domain $[0, \sqrt{\kappa_0}]$ are strongly blue-shifted by amount $\sim h^{1/3}$, leaving a gap near $\omega = 0$. Consequently, as stated above, we observe that in the absence of interactions a gapless density of states cannot possibly emerge. The only possible scenario in which Eq. (2) with $\kappa_{\text{shift}}(h, J, \kappa_0) = 0$ can lead to a gapless density of states is that $f(h, J, \kappa_0)$ cancels $h$. This can happen despite $J$ being small, because $f(h, J, \kappa_0)$ is a random variable that can experience large fluctuations. Since $f$ is normally distributed, the probability to observe a $f = -h$ fluctuation is given by $(J \sqrt{\langle x_*^2 \rangle})^{-1} \exp[-h^2 / (2 J^2 \langle x_*^2 \rangle)]$. Using the scaling

prediction derived above, $\langle x_*^2 \rangle \sim h^{4/3}/\kappa_0$, and recalling that $y = J/(h^{1/3}\kappa_0^{1/2})$, we conclude that $\log[y\,g(y)] \sim -y^{-2}$.

With the dimensionless function $g(y)$ at hand, we use the scaling prediction $A_g(h,J,\kappa_0) = g(y)/(\kappa_0 h)$ to obtain

$$\log\left[\kappa_0^{1/2} h^{2/3} J A_g\right] \sim -\frac{\kappa_0 h^{2/3}}{J^2}\,, \tag{3}$$

in the weak interactions regime. This prediction is tested against extensive numerical data in Fig. 4, revealing excellent quantitative agreement. The predominantly exponential variation of $A_g(h,J,\kappa_0)$ with $-\kappa_0 h^{2/3}/J^2$ is reminiscent of the predominantly exponential variation of $A_g$ with $-1/T_p$ in computer glasses [17]. This similarity is suggestive, calling for a better understanding of the possible relations between the model parameters $\kappa_0$, $h$ and $J$, and the parent temperature $T_p$ that characterizes the liquid state at which the glass falls out of equilibrium during a quench. This interesting issue is further discussed below.

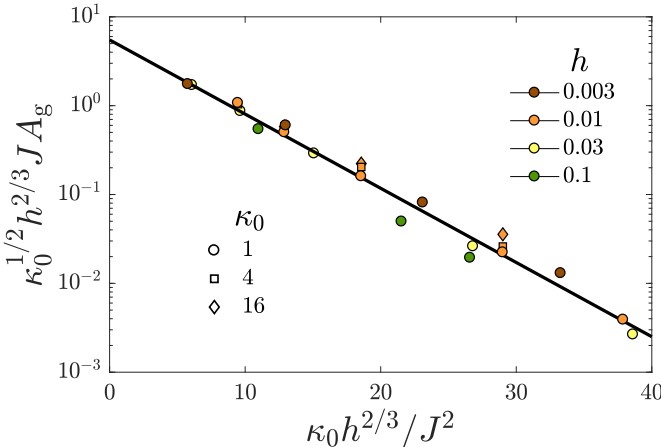

Figure 4: Validation of the weak-interaction regime prediction given by Eq. (3) for the prefactor $A_g(h,J,\kappa_0)$ of the $\sim \omega^4$ glassy spectrum, for various values of the parameters $J, \kappa_0$ and $h$ as indicated by the legend.

## 4 The intermediate-strength interactions regime

What happens when the interaction strength $J$ is further increased, beyond the weak interactions regime? How can we properly define the two regimes and the transition between them? To address these questions, we need to consider again the effective potential of Eq. (2). In the weak interactions regime, $\omega_\times(h) \sim h^{1/3}$ is a central frequency scale set by $h$ alone, and perturbative corrections for small $J$ have been considered. The latter are mainly related to the effective random force $f(h,J,\kappa_0)$ in Eq. (2), while the stiffness shift $\kappa_{\text{shift}}(h,J,\kappa_0)$ is negligible. As $J$ is increased, we expect to enter a regime where the constant force $h$ plays no role anymore, but where the stiffness shift $\kappa_{\text{shift}}$ plays a dominant role. To see this, consider Eq. (2) with $h = 0$; assuming that large fluctuations in the effective random force $f$ play no major role here, we immediately observe that 'small' and 'large' $\kappa$ is defined relative to the stiffness shift $\kappa_{\text{shift}}$. Consequently, we identify the latter as $\omega_\times^2$ in this regime, which we term the intermediate-strength interactions regime. Moreover, we expect $\omega_\times = \sqrt{\kappa_{\text{shift}}}$ in this regime to be a function of $J$ and $\kappa_0$ (as $h$ is expected to play no role here), and the crossover between the weak and intermediate strength regimes to be determined by the interaction strength $J$ for

which the to-be-calculated $\omega_\times(J, \kappa_0)$ smoothly connects to $h^{1/3}$, the prediction for $\omega_\times$ in the weak interaction regime.

While the calculation of $\omega_\times^2 = \kappa_{\text{shift}}$ in the intermediate-strength interactions regime goes beyond a scaling theory and requires additional analysis, the scaling theory provides strong predictions for the role of $\omega_\times$ in the intermediate-strength interactions regime, and hence allows to cleanly determine it numerically. To see this, we consider the mean square displacement $\langle x_*^2 \rangle$, by analyzing Eq. (2) with $h = 0$, closely following the derivation presented above in the weak interactions regime. For $\kappa \ll \kappa_{\text{shift}} = \omega_\times^2$, $\kappa$ can be neglected and we obtain $\langle \tilde{x}_*^2 \rangle(\kappa) \sim \omega_\times^2$. For $\kappa \gg \kappa_{\text{shift}} = \omega_\times^2$, $\kappa_{\text{shift}}$ can be neglected and we obtain $\langle \tilde{x}_*^2 \rangle(\kappa) \sim \kappa^{-2}$. Evaluating again the partial average $\langle \bar{x}_*^2 \rangle(\kappa) = \kappa^{-1} \int_0^\kappa \langle \tilde{x}_*^2 \rangle(\kappa') d\kappa'$ (such that $\langle x_*^2 \rangle = \langle \bar{x}_*^2 \rangle(\kappa_0)$ and recall that $p(\kappa) = \kappa_0^{-1}$), we obtain $\langle \bar{x}_*^2 \rangle(\kappa) \sim \omega_\times^2$ for $\sqrt{\kappa} \ll \omega_\times(J, \kappa_0)$, while for $\sqrt{\kappa} \gg \omega_\times(J, \kappa_0)$ we have $\langle \bar{x}_*^2 \rangle(\kappa) \sim \omega_\times^4 / \kappa$, where the amplitude of the latter has been set such that the two scaling laws smoothly connect for $\sqrt{\kappa} \approx \omega_\times(J, \kappa_0)$. Note that all of these results, once presented in terms of $\omega_\times$, are identical to the corresponding results in the weak interactions regime, yet again highlighting the central role played by the characteristic frequency $\omega_\times$ in the KHGPS model.

The above analysis predicts that $\langle \bar{x}_*^2 \rangle(\kappa) / \omega_\times^2$ is a function of $\kappa / \omega_\times^2$ that is independent of $J$ and $\kappa_0$, once $\omega_\times(J, \kappa_0)$ is properly identified. Hence, we can ask what function $\omega_\times(J, \kappa_0)$ generates the predicted collapse, and determine it numerically. This procedure is presented in Fig. 5a, where both a perfect collapse is demonstrated and $\omega_\times(J, \kappa_0)$ is extracted (inset). The latter shows that $\omega_\times(J, \kappa_0)$ is a predominantly power-law (the numerical data suggest the power-law approximation $\omega_\times(J, \kappa_0) \sim J^{7/8} / \kappa_0^{3/8}$, indicated by the straight line in the inset, added as a guide to the eye). The success of the $\langle \bar{x}_*^2 \rangle$ analysis then implies $\langle x_*^2 \rangle = \langle \bar{x}_*^2 \rangle(\kappa_0) \sim \omega_\times^4 / \kappa_0$. Furthermore, with the numerical $\omega_\times(J, \kappa_0)$ at hand, we can determine the crossover $J_\times(h, \kappa_0)$ between the weak and intermediate-strength interactions regimes by numerically solving $\omega_\times(J_\times, \kappa_0) \sim h^{1/3}$ (note that the numerical power-law approximation for $\omega_\times(J, \kappa_0)$ implies $J_\times(h, \kappa_0) \sim h^{8/21} \kappa_0^{3/7}$). The complete scaling predictions for $\langle x_*^2 \rangle$, in both the weak interactions and intermediate-strength interactions regimes and including the crossover interaction strength $J_\times(h, \kappa_0)$, are verified against extensive numerical simulations in Fig. 5b.

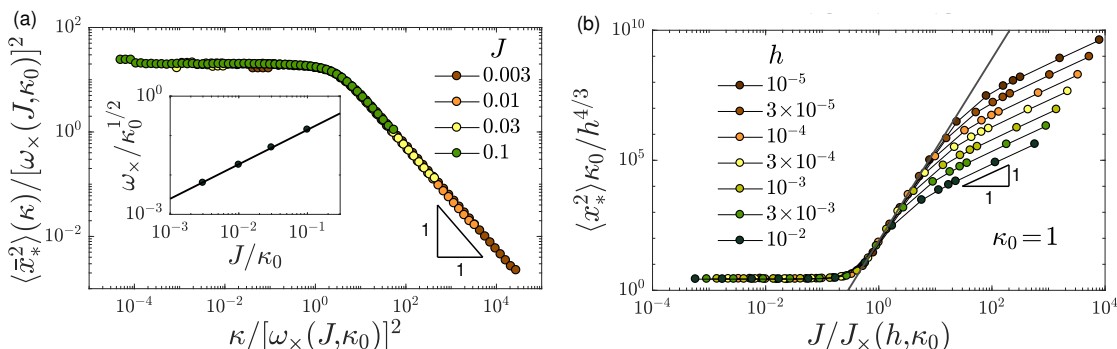

Figure 5: (a) We extract the function $\omega_\times(J)$ in the intermediate-strength interactions regime, by choosing values that lead to a collapse of the partial averages $\langle \bar{x}_*^2 \rangle(\kappa)$ when rescaled by $\omega_\times^2$ and plotted against $\kappa / \omega_\times^2$, with a crossover at $\kappa / \omega_\times^2 \sim \mathcal{O}(1)$. We find numerically that $J^{7/8}$ represents a good analytic approximation for the crossover frequency $\omega_\times(J)$, see inset. (b) The mean squared displacements $\langle x_*^2 \rangle$ of oscillator coordinates are shown to follow the prediction $\langle x_*^2 \rangle \sim \omega_\times^4 / \kappa_0$, where $\omega_\times(J, h, \kappa_0)$ follows different scaling laws in the weak- $(J < J_\times)$ and intermediate- $(J > J_\times)$ strength interaction regimes, as shown above. In the $J \gg \kappa_0$ regime at which interactions dominate, we observe a trivial $\langle x_*^2 \rangle \sim J$ scaling behavior.

Finally, we consider the prefactor $A_g$. Repeating the derivation detailed above in the weak interaction regime verbatim, we obtain $A_g(h, J, \kappa_0) = s(z)/(\kappa_0[\omega_\times(J, \kappa_0)]^3)$, where $s(z)$ is a dimensionless function of the dimensionless quantity $z \equiv J/\kappa_0 \ll 1$, which formally cannot be obtained by pure scaling considerations. However, the analysis suggests that unlike the weak interactions regime (where a strongly varying multiplicative dimensionless function $f(y) \ll 1$ exists, cf. Eq. (3)), there exists no additional strong dependence on $z$ in the intermediate-strength interactions regime (of course we cannot exclude the possibility that it is a very weak function of $z$). Consequently we take $s(z)$ to be a constant (which is expected to be of order unity) to obtain

$$A_g(h, J, \kappa_0) \sim \frac{1}{\kappa_0[\omega_\times(J, \kappa_0)]^3} \qquad \text{for} \qquad J \gg J_\times(h, \kappa_0) \,, \qquad (4)$$

where $\omega_\times(J, \kappa_0)$ has been numerically obtained in the inset of Fig. 5a. Using the latter, the prediction in Eq. (4) is verified against extensive numerical simulations in Fig. 6. Since $\omega_\times(J, \kappa_0)$ is numerically approximated by a power-law, so is $A_g(h, J, \kappa_0)$, and we conclude that $A_g$ decays predominantly as a power-law in the intermediate-strength interactions regime (within the numerical power-law approximation, we have $A_g(h, J, \kappa_0) \sim J^{-21/8}/\kappa_0^{-1/8}$, which corresponds to the line shown in Fig. 6).

Now that we have the crossover interaction strength $J_\times(h, \kappa_0)$ at hand, it is clear that $A_g(h, J, \kappa_0)$ in Eq. (3) is in fact valid for $J \ll J_\times(h, \kappa_0)$. Consequently, $A_g(h, J, \kappa_0)$ is a non-monotonic function of the interaction strength $J$, increasing with it for $J \ll J_\times(h, \kappa_0)$ (as described by Eq. (3)) and decreasing with it for $J \gg J_\times(h, \kappa_0)$ (as described by Eq. (4)), with markedly different functional forms. This non-monotonic dependence on $J$ is explicitly demonstrated in Fig. 6. The intermediate-strength interactions regime clearly crosses over to yet another regime at $J \sim \kappa_0$, i.e. when $z = J/\kappa_0$ is no longer small. The strong interactions regime, $J \gg \kappa_0$, is not discussed here, though the expected scaling relation $\langle x_*^2 \rangle \sim J$ is in fact observed in Fig. 5b.

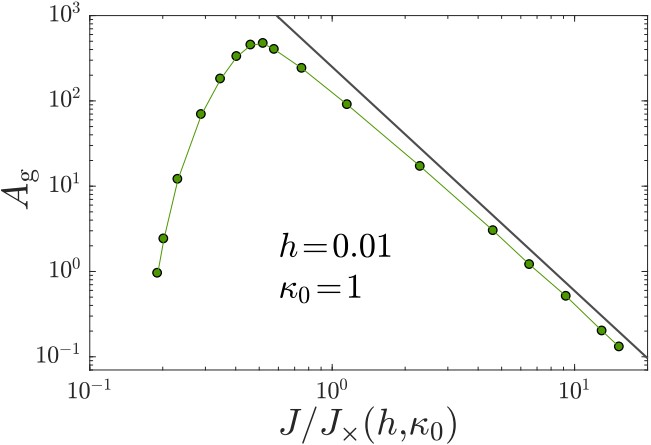

Figure 6: We estimate the prefactors $A_g$ for $h = 0.01$, $\kappa_0 = 1$, and various values of $J$, via the numerically calculated $\mathcal{D}(\omega)$ as shown e.g. in Fig. 2. As predicted, $A_g$ varies non-monotonically, peaking at $J/J_\times \sim \mathcal{O}(1)$. The solid line corresponds to $\kappa_0^{-1}\omega_\times^{-3}$, where $\omega_\times(J, \kappa_0)$ was extracted as explained in Fig. 5.

# 5   Conclusion

In this work, we introduced and studied a mean-field model of interacting quasilocalized excitations, termed the KHGPS model and defined by the Hamiltonian in Eq. (1). A major result of our analysis is that local minima of the model appear to robustly feature $\mathcal{D}(\omega) \sim \omega^4$ vibrational spectra, independently of the input parameters $h$, $J$ and $\kappa_0$. As such, the KHGPS Hamiltonian in Eq. (1) appears to offer a relatively simple model for the emergence of the universal $\mathcal{D}(\omega) \sim \omega^4$ nonphononic spectra, previously observed in finite-dimensional, particle-based computer glass-formers [10–15, 18].

Several other non-mean-field models [22, 23] and phenomenological theories [1, 24–26] were previously put forward to the same aim; most of them, however, require parameter fine-tuning [22–24, 26] or some rather strong a priori assumptions [1]. In addition, several other mean-field models, introduced in order to explain the low-frequency spectra of structural glasses, predict $\mathcal{D}(\omega) \sim \omega^2$, independently of spatial dimension [16, 39–41]. In light of these previous efforts, our results appear to support — and further highlight — GPS's suggestion [19, 20] that stabilizing anharmonicities — absent from the aforementioned mean-field models — constitute a necessary physical ingredient for observing the universal $\sim \omega^4$ law in this class of mean-field models.

We also developed a comprehensive scaling theory of the salient quantities in the model — the oscillators mean square displacement $\langle x_*^2 \rangle(h, J, \kappa_0)$, the emergent characteristic frequency scale $\omega_\times(h, J, \kappa_0)$ and the non-universal prefactor $A_g(h, J, \kappa_0)$ — and supported the theoretical predictions by extensive numerical simulations. Our results show that the internal force $h$, which is absent from GPS's original work [19, 20], is responsible for the existence of two distinct regimes, where the $\omega^4$ law emerges from quite different ingredients. One regime is characterized by weak inter-oscillator interactions, where $h$ plays an important role, and the other by stronger interactions, where $h$ plays no role. In both regimes the frequency scale $\omega_\times(h, J, \kappa_0)$ plays important roles, but in the former regime large fluctuations in the inter-oscillator interactions result in a predominantly exponential dependence of $A_g(h, J, \kappa_0)$ on $-(\kappa_0 h^{2/3}/J^2)$. The latter is reminiscent of recent observations in computer glasses, where the control parameter is the temperature $T_p$ at which a glass falls out of equilibrium (cf. inset of Fig. 1a), and hence might offer a promising route to link the model to realistic glass formation processes.

Our findings give rise to several interesting questions and research directions. First, while we provide strong numerical support to the generic emergence of the $\mathcal{D}(\omega) \sim \omega^4$ density of states in the KHGPS model, it would be desirable to obtain a rigorous proof of this observation and its validity conditions [42]. Second, it would be most useful to further explore the analogy between the KHGPS model and finite-dimensional glasses, better understanding the hypothesized relation between the minimization of the Hamiltonian and the self-organization processes taking place while a glass is quenched from a melt. In particular, it would be interesting to understand whether and how the model parameters $h$, $J$ and $\kappa_0$ might be related to measurable quantities in supercooled liquids and glasses.

Finally, in the analysis above, the stiffness scale $\kappa_0$ fully characterized the initial stiffness distribution $p(\kappa)$ — describing the non-interacting oscillators — that was taken to be gapless and uniform in the interval $[0, \kappa_0]$. It would very interesting to understand the relations between $p(\kappa)$ and liquid states above the glass temperature, both in terms of its functional form and in relation to the possible existence of a gap in it [42], which has not been considered here. Establishing such relations may clarify what mean-field models such as the KHGPS one can teach us about the physics of glasses.

# Acknowledgements

We benefited from discussions with Giulio Biroli, Jean-Philippe Bouchaud, Gustavo Düring, Eric De Giuli, and Guilhem Semerjian.

**Funding information**   This project has received funding from the European Research Council (ERC) under the European Union's Horizon 2020 research and innovation programme (grant agreement no. 723955 - GlassUniversality) and by a grant from the Simons Foundation (#454955, Francesco Zamponi). P. U. acknowledges support by "Investissements d'Avenir" LabEx-PALM (ANR-10-LABX-0039-PALM). E. L. acknowledges support from the NWO (Vidi grant no. 680-47-554/3259). E. B. acknowledges support from the Minerva Foundation with funding from the Federal German Ministry for Education and Research, the Ben May Center for Chemical Theory and Computation and the Harold Perlman Family.

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
