# Peer review of "Mean-field model of interacting quasilocalized excitations in glasses"

_SciPost Physics Core, doi:SciPost Phys. Core 4, 008 (2021)_

## Round 1 · Referee Report · Anonymous (Referee 1) · 2021-2-18

Report

In this paper the properties of a model introduced by Kühn and Horstmann and Gurevich et al. describing local defects in glasses are investigated. The model consists of a set or random anharmonic oscillators, coupled linearly to each other in the presence of an external force h. The randomness is given in terms of spatially fluctuating harmonic stiffness coefficients and couplings. The distribution density of the stiffness coefficients is assumed to be uniform and extend to zero. The couplings are assumed to fluctuate according to a Gaussian of variance $\sim J^2$.

By means of the combination of a scaling analysis and numerical simulations the authors establish the following results:

  • There is always a low-frequency density of states (DOS) proportional to the fourth power of frequency
  • For weak interaction strength $J$ the crossover to a linear behavior of the DOS occurs at a frequency $w_x$, which is proportional to $h^3/2$, for larger $J$, $w_x$ becomes independent of $h$ and is related to the effective mean shift of the oscillator stiffness due to the anharmonicity.
  • These findings lead to a strong non-monotonic dependence of the prefactor of the $w^4$ DOS on the interaction strength $J$: in the weak-interaction regime it increases exponentially and then crosses over to an inverse-power law.

The presentation, the results and the discussion are highly interesting and important for the glass community, in which the role of quasilocalized oscillaters has been widely debated recently.

The paper would benefit from proof-reading by a native-English speaking person.

---

## Round 1 · Referee Report · Anonymous (Referee 2) · 2021-3-15

Strengths

This paper presents a mean field analysis of the low frequency of a density of states for a simple Hamiltonian designed to model local anharmonic vibrations that are pairwise coupled. The analysis recovers a quartic power law in frequency, similar to that observed in simulations of more realistic model glasses. Numerical calculations establish the dependence of the absolute magnitude of the density of states on the coupling strength J. This is a valuable contribution to the study of the novel low frequency mechanical properties of glassy solids.

Weaknesses

The coupling between these localized anharmonicities seemed poorly rationalized, given its significance. I was left to wonder about the more obvious coupling between each individual anharmonicity and the surrounding elastic medium. I can imagine that this coupling may not contribute significantly at low frequencies but some discussion on this point would be helpful.

Report

I recommend the paper be accepted for publication once the authors have had a chance to consider the comment above.

---

## Editorial Decision

published